# Computational Biomechanics of Sleep: A Systematic Mapping Review

**DOI:** 10.3390/bioengineering10080917

**Published:** 2023-08-02

**Authors:** Ethan Shiu-Wang Cheng, Derek Ka-Hei Lai, Ye-Jiao Mao, Timothy Tin-Yan Lee, Wing-Kai Lam, James Chung-Wai Cheung, Duo Wai-Chi Wong

**Affiliations:** 1Department of Biomedical Engineering, Faculty of Engineering, The Hong Kong Polytechnic University, Hong Kong; 2Department of Electronic and Information Engineering, Faculty of Engineering, The Hong Kong Polytechnic University, Hong Kong; 3Sports Information and External Affairs Centre, Hong Kong Sports Institute, Hong Kong; 4Research Institute for Sports Science and Technology, The Hong Kong Polytechnic University, Hong Kong

**Keywords:** finite element, in silico, computer experiment, musculoskeletal pain, insomnia, spine alignment

## Abstract

Biomechanical studies play an important role in understanding the pathophysiology of sleep disorders and providing insights to maintain sleep health. Computational methods facilitate a versatile platform to analyze various biomechanical factors in silico, which would otherwise be difficult through in vivo experiments. The objective of this review is to examine and map the applications of computational biomechanics to sleep-related research topics, including sleep medicine and sleep ergonomics. A systematic search was conducted on PubMed, Scopus, and Web of Science. Research gaps were identified through data synthesis on variants, outcomes, and highlighted features, as well as evidence maps on basic modeling considerations and modeling components of the eligible studies. Twenty-seven studies (*n* = 27) were categorized into sleep ergonomics (*n* = 2 on pillow; *n* = 3 on mattress), sleep-related breathing disorders (*n* = 19 on obstructive sleep apnea), and sleep-related movement disorders (*n* = 3 on sleep bruxism). The effects of pillow height and mattress stiffness on spinal curvature were explored. Stress on the temporomandibular joint, and therefore its disorder, was the primary focus of investigations on sleep bruxism. Using finite element morphometry and fluid–structure interaction, studies on obstructive sleep apnea investigated the effects of anatomical variations, muscle activation of the tongue and soft palate, and gravitational direction on the collapse and blockade of the upper airway, in addition to the airflow pressure distribution. Model validation has been one of the greatest hurdles, while single-subject design and surrogate techniques have led to concerns about external validity. Future research might endeavor to reconstruct patient-specific models with patient-specific loading profiles in a larger cohort. Studies on sleep ergonomics research may pave the way for determining ideal spine curvature, in addition to simulating side-lying sleep postures. Sleep bruxism studies may analyze the accumulated dental damage and wear. Research on OSA treatments using computational approaches warrants further investigation.

## 1. Introduction

Sleep problems or deprivations are a widespread condition affecting millions of people worldwide, ranging from as mild as snoring and a restless night [1] to as serious as causing stroke [2] and heart failure [3]. Existing studies showed that sleep deprivation has significant negative effects on health, productivity, and quality of life. More than one-third of the population reported insomnia complaints [4], and almost one billion of people worldwide are afflicted by sleep apnea, with prevalence in certain counties over 50% [5]. They are significant risk factors for a variety of chronic conditions, such as hypertension, asthma, and dementia [6,7]. People with insomnia were 2.5 times more likely to have several chronic conditions [8]. Sleep-deprived drivers are twice as likely to be involved in an accident, endangering others [9]. One-fifth and one-tenth, respectively, of those who suffered from sleep deprivation reported at least one home accident and one work accident in the past 12 months [10]. In addition, the burden associated with sleep problems is increasing due to COVID-19 [11], the aging population [12], the hectic lifestyle in modern society [13], and the use of technological gadgets [14].

Sleep medicine encompasses a broad range of fields, including psychiatric, neurological, cardiopulmonary, and otolaryngological [15]. Biomechanics play an important role, directly or indirectly, in these areas regarding some of the sleep issues. For instance, sleep quality and comfort are intertwined with insomnia [16] and are related to the design of beddings, particularly in terms of musculoskeletal (dis)comfort [17,18,19]. Research on sleep ergonomics focuses on the shape and material designs of beds, mattresses, pillows, and other bedding components to improve sleep quality and sleep health [17,20,21]. Specifically, the design shall facilitate a desirable spine curvature, preventing aberrant twisting and sagging of the neck and trunk, and consequently musculoskeletal pain and issues [17,20,21]. The pressure of body–mattress contact is another indicator of tactile comfort [20]. Concentrated pressure and stress resulting from poor bedding design may induce tissue damage and poor localized blood supply, hence increasing the risk of pressure ulcers in bed-bound patients [22].

Moreover, biomechanical studies offer valuable insights into the pathoanatomical and pathobiomechanical components that contribute to various sleep disorders. For instance, the collapse of the upper airway during sleep is a primary cause of sleep apnea and snoring, which can be caused by issues with the soft palate, tongue, hyoid bone, pharyngeal walls, and muscles [15,23]. By studying the biomechanics of the upper airway, researchers were able to gain a better understanding of the source of the collapse and devise effective measures. Barbero et al. [24] evaluated how the positioning of a mandibular advancement device could better improve the upper airway volume and inspiratory pressure gradient, whereas Yoon et al. [25] and Zhiguo et al. [26] aimed to quantify the abnormal stress and thus adverse effects of mandibular advancement devices on teeth and periodontal ligaments and pave the way for better design.

Nevertheless, traditional biomechanical studies in vivo were limited by the practicability of measurements and could only be undertaken on the exterior of the body or in areas where sensors could be placed, unless cadaveric or animal studies were conducted. The strength of computational biomechanics in silico lies in its ability to facilitate a versatile platform for investigating the biomechanics of the body’s internal structure in a controlled environment with pre-assigned conditions and evaluating the sensitivity of different intrinsic or extrinsic factors to support design and clinical applications [27]. Computational modeling approaches employing the finite element (FE) method enable the modeling of irregular multiple geometries, complex material properties and loading conditions to explore pathomechanisms, predict treatment outcomes, and support designs [28,29,30,31,32,33], which has been widely utilized in the fields of orthopedics [28,30,31], dentistry [34,35], pulmonary [32], otolaryngology [36], and design ergonomics [29,37,38]. The FE method works by splitting complex and irregular geometries into smaller and simpler components called FEs that are connected by a mesh, then predicting the mechanical responses towards a set of pre-assigned material properties and boundary and loading conditions on each element using mathematical equations.

To this end, the objective of this study is to conduct a structural literature search with evidence maps on the application of FE (i.e., computational biomechanics) on sleep-related topics. The goal is to provide succinct information on the basic considerations and configurations in these in silico studies and how their outcomes might contribute to a better understanding of the biomechanics aspects of sleep. This review delimited the technical details in modeling, simulation, and validation, as well as the scope of treatment interventions.

## 2. Review Methodology

### 2.1. Search Strategy

The search terms were predetermined by simple pilot searches utilizing keywords or free-text words [39]. The search terms comprised “finite element” AND sleep-related terms, including “sleep*”, “snor*”, “recumben*”, “insomnia*”, “dyssomnia*”, “parasomnia*”, “hypersomnia*”, “somn*”, “restless”, “bed*”, “pillow*”, “mattress*”, “topper*”, “blanket*”, and “quilt*”.

Since we noticed a lot of incorrect results from the geology, chemical engineering, and structural engineering disciplines, we used the (AND) NOT operator on the following phrases to reduce the number of inaccurate searches, in addition to keywords related to exclusion criteria. They included “sit*”, “seat*”, “impact”, “amputee*”, “prosthe*”, “dressing”, “tumor*”, “tumour*”, “cancer*”, “surger*”, “implant*”, “screw*”, “drug*”, “fracture*”, “replacement*”, “alloy*”, “superalloy*”, “railway”, “transport*”, “turbine*”, “machin*”, “sediment*’, “soil*”, “sand*”, “rock*”, “vehicle*”, “bed fusion”, “electrochemical”, “somnodent”, “thermos*”, “temperature*”, “lattice*”, and “dielectric*”.

The first and second authors (E.S.-W.C. and D.K.-H.L.) independently searched the literature in mid-June 2023 from three major electronic databases, including PubMed (field: title/abstract, filter: English), Web of Science (field: topic, filter: English, journal articles), and Scopus (field: title/abstract/keywords, filter: English). After the search and screening process from the aforementioned databases, a grey literature search on Google Scholar was conducted by the third author (Y.-J.M.) using “finite element” and terms related to sleep-related disorders and their anatomical sites. We only selected original journal articles published in journals. There was no limitation on the year of publication.

### 2.2. Eligibility Criteria and Screening Process

The scope of the review included studies exploiting FE methods to explore the biomechanics of sleep-related topics, particularly sleep medicine. Sleep ergonomics was also included because we viewed it as the “biomechanics” of sleep disorders, such as insomnia and sleep-related musculoskeletal pain [18]. Nevertheless, we decided not to include treatments and interventions of sleep medicine (e.g., surgery, implant, and sleep position training) in this mapping review.

Inclusion criteria included: (1) journal articles (including preprints and early access articles); (2) published in English; (3) original research; (4) articles that investigated and/or clearly implicated (in title and/or abstract) the biomechanics of sleep behavior/posture/recumbence, sleep medicine, and sleep ergonomics; and (5) utilized FE method.

Exclusion criteria included: (1) non-journal articles (e.g., conference papers, book sections, and thesis); (2) non-original research (e.g., review and perspective papers); (3) did not model or model from any human body parts; (4) models from specific populations, including amputation, surgery, fractures, traumatic injuries, or tumor; and (5) FE simulations in thermodynamics, acoustics/vibratory, electrostatic, and magnetic field phase domains.

### 2.3. Data Extraction and Synthesis

Apart from sleep ergonomics, the areas of interest would be classified according to the International Classification of Sleep Disorder (ICSD), third edition [40], into six categories: insomnia, sleep-related breathing disorders, hypersomnolence disorders, circadian rhythm sleep–wake disorders, parasomnia, and sleep-related movement disorders. An evidence map, with the categories, would be constructed with the publication years, and the basic setting of the computational studies using the Sankey diagram. Data synthesis would be conducted on the simulating case scenarios, variants, and outcomes, in addition to the highlighted features of the studies, where applicable. Another evidence map would be produced to lay out the model components via a dot-matrix plot, and specific considerations under each category.

## 3. Study Selection

As shown in Figure 1, the initial search identified 1852 studies, and 1208 records were screened after removing 644 duplicates. We did a primary screening by title, abstract, and keywords, which resulted in the exclusion of 1080 publications for various reasons (violating the publication type criteria, *n* = 16; violating the article type criteria, *n* = 4; irrelevant to the review scope, *n* = 1060). Then, the eligibility of the 129 articles was evaluated through the full texts, and 105 studies were excluded for not investigating or directly implicating sleep-related topics, *n* = 48; not applying FE methods, *n* = 9; violating phase domain requirement, *n* = 2; modeling of animals, *n* = 8; did not model or model from any body parts, *n* = 7; treatment interventions, *n* = 10; pressure ulcers/sores, *n* = 19. In addition, two contentious studies were excluded upon consensus (others, *n* = 2). The first focused on the microgravity [41], while the second one targeted on Down syndrome [42]. A Google Scholar search for grey literature yielded four results, totaling 27 studies conclusive for qualitative synthesis in this review [43,44,45,46,47,48,49,50,51,52,53,54,55,56,57,58,59,60,61,62,63,64,65,66,67,68,69,70].

## 4. Overview and Evidence Mapping

Figure 2 and Figure 3 map the general classifications and contexts of the included research. Three of the included studies (*n* = 3, 11.1%) focused on sleep bruxism, a kind of sleep-related movement disorder. Five studies (*n* = 5, 18.5%) investigated the sleep ergonomics of pillows (*n* = 2) and mattresses (*n* = 3). The remaining articles (*n* = 19, 70.3%) dealt with sleep-related breathing disorders, focusing solely on obstructive sleep apnea (OSA). There were no studies directly related to the other four categories, including insomnia, sleep-hypersomnolence disorders, circadian rhythm sleep–wake disorders, and parasomnia.

The earliest article dates back two decades (year range of included studies: 2002–2022). Despite working with two-dimensional (2D) models, the team was able to simulate fluid–structure interaction (FSI), which was relatively computationally intensive. There were 7 and 20 studies, respectively, that reconstructed 2D and three-dimensional (3D) models. For studies that did not explicitly address the FE solver, a solid mechanics dynamic solver was assumed for adaptive response modeling, especially neuromuscular control. Except for one study, all sleep ergonomic investigations used a solid-static/quasi-static solver, while all bruxism studies used a dynamic solver. The majority of OSA investigations applied FSI (*n* = 9); six of them (*n* = 5 dynamics; *n* = 1 static/quasi-static) employed solid mechanics modeling, while four of them exploited FE morphometry (*n* = 5).

More than half of the studies (*n* = 17, 63.0%) took a single-subject, subject-specific approach. Six studies (*n* = 6, 22.2%) explored a case series, with two and four of them, respectively, focusing on sleep ergonomics and OSA. FE morphometry studies resumed a cross-sectional research design with a range of total sample sizes from 38 to 108. Nevertheless, they were all 2D models, which implied that 3D models in a large cohort might not be easily realizable.

## 5. Thematic Analysis

### 5.1. Sleep Bruxism

Sleep bruxism is a sleep-related movement disorder characterized by involuntary clenching and grinding of the teeth during sleep [71]. It was reported that more than 30% of children reported signs of sleep bruxism [72], and the prevalence of adults was 16.5% in the Netherlands [73]. In addition to dental wear, sleep bruxism might induce temporomandibular joint disorder, which can result in headaches, earaches, and facial pain [71].

Therefore, the three included studies all strive to investigate the stress on the articular disc, which has been associated with the temporomandibular joint disorder (Table 1 and Figure 4a). Commisso et al. [50] compared sustained and cyclic clenching with different rates and magnitudes of muscle activation. They predicted that sustained clenching caused more shear stress on the articular disc and was potentially more harmful than cyclic clenching. Considering the dynamic modeling of the loading and clenching behavior, non-linear viscoelasticity appeared to be crucial in the simulation. Nonetheless, Commisso et al. [50] simplified the model considerably by excluding the upper jaw and the temporal bone and constraining the teeth to replicate the chewing action.

The other two studies conducted by Sagl’s research team [66,67] involved extensive modeling work and have been referenced in terms of model creation and validation [74,75,76], as shown in Figure 4. In addition to comprehensive modeling on the temporomandibular (TMJ) ligaments and muscle load (Figure 4 and Figure 5a), they simulated lateral bruxing and examined the impact of various parameters, including tooth morphology, position, grinding direction, and bruxing power [66,67]. They emphasized their novelty in the estimation of muscle force by the forward-dynamics tracking approach that incorporated a reaction force tracking [77]. In addition to the stress at the temporomandibular joint, the authors explored the risk of dental wear by predicting the bruxing force at the molar and canine with computer models [66].

**Figure 4 bioengineering-10-00917-f004:**
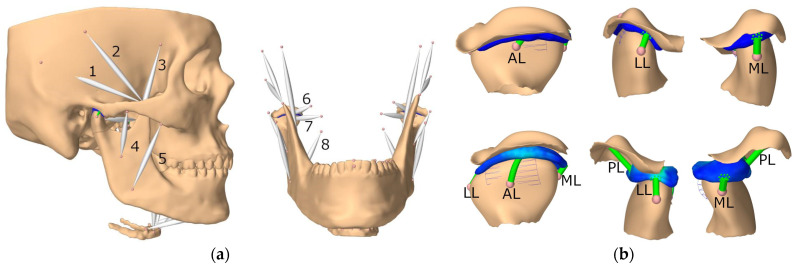
Model reconstruction of the skull and mandible illustrating: (**a**) the muscles and (**b**) temporomandibular joint (TMJ) ligaments. 1, posterior temporalis; 2, medial temporalis; 3, anterior temporalis; 4, deep master; 5, superficial master; 6, superior head of lateral pterygoid; 7, inferior head of lateral pterygoid; 8, medial pterygoid; AL, anterior ligament; LL, lateral ligament; ML, medial ligament; PL, posterior ligament (source: [76], under Creative Commons Attribution License).

### 5.2. Sleep Ergonomics

Sleep ergonomics is closely related to insomnia [20,78]. Poorly designed beddings might produce discomfort, pain, and improper sleep postures, making it difficult for individuals to fall asleep and remain asleep [17,20]. Due to the high demand in computing for the simulation of the entire human body, computational modeling of sleep ergonomics in bedding design remains a great challenge [51,70]. In light of this, two studies simplified the work into two-dimensional models while developing three models [51,70]. One study reconstructed the encapsulated soft tissue of the head–neck complex from an individual at the 50th percentile from a random sample of 63 adult females [49]. Recent research aimed to develop subject-specific, comprehensive, anatomically detailed, 3D models [53,64], as shown in Figure 6.

Among the five studies in this category, two investigated the designs of pillows [49,64], while three targeted the designs of mattresses (Table 2 and Figure 5b). All studies only considered the supine lying posture. Using elevator mats, different pillow heights of a B-shaped pillow (often referred to as a cervical pillow) were simulated and examined. Ren et al. [64] simulated 110, 130, 150, and 170 mm pillow heights (total height). They reported extension and lordosis trends of the neck with increasing pillow heights but have not indicated which pillow height is more advantageous. Chen et al. [49] designed a B-shaped pillow with two peak heights at 48 and 65 mm, based on the head–neck curvature during standing, and simulated elevations of 10, 20, 30, 40, and 50 mm. Chen et al. [49] concluded that an overall elevation of 35.9 mm (based on regression) could be the optimal pillow height since it closely approximated the same head–neck curvature following compression. Nevertheless, they reported a skeptical V-shaped trend on the stress of the neck. We believed that this may be owing to the unconstrained inferior cross-section of the neck, which shall be pulled by the spine from the trunk, as well as the use of a dynamic explicit solver for static scenarios. This issue may not exist in the model reconstructed by Ren et al. [64] since their model recreated the cervical spine, rib cage, and thoracic region with simulation resting on both the pillow and mattress.

On the other hand, mattress stiffness was the primary design consideration. Hong et al. [53] tested soft, medium, and hard mattresses (20, 42, and 120 lbs at 25% compression indentation load defection, respectively) on spinal curvature, contact pressure, and intervertebral disc stress, while featuring the model validation with unobstructed measurement on spinal curvature [79]. In addition, Yoshida et al. [70] evaluated pocket coil mattresses with stiffnesses of 6.0, 9.6, 11.4, and 14.0 kPa, examined the von Mises stress (VMS) of the cervical spine and the sinking displacement of the head and chest, and linked to the perceived comfort and preference of the individuals. However, without the consideration of a pillow model, we questioned the significance and implications of assessing cervical spine stress. To achieve optimal support specific to body regions, Denninger et al. [51] designed a foam mattress with cubic cells of varying cavity sizes and shapes to provide customized support to various body locations. The two-dimensional simulation was made possible by reducing the human body model to a series of juxtaposed ellipsoid layers and determining the sinkage of the spine at L4 to C7 levels for different cubic cell designs and arrangements.

### 5.3. Obstructive Sleep Apnea (OSA)

OSA is often characterized by the collapse and obstruction of the upper airway recurrently during sleep with a complex and debated pathophysiology [80]. Abnormalities of anatomic structures might result in constriction of the upper airway, whereas diminished neural control or abnormalities of the dilatation muscles may compromise airway patency [48,80]. Some studies believed that airway collapse occurred during inhalation due to an increase in negative pressure [81,82], while others felt that it happened at the end of the exhalation phase due to a reduction in retropharyngeal volume and an increase in soft palate movement [83,84]. Alternatively, OSA may occur in people with hypertrophic or flaccid soft palates [48]. Among the 19 studies related to OSA, 4 studies (*n* = 4) applied FE morphometry, 6 studies (*n* = 6) utilized the standard FE analysis on solid mechanics, and 9 studies (*n* = 9) performed co-simulation on FSI. The data and evidence map on the modeling are given in Table 3, Table 4 and Table 5 and Figure 7.

FE morphometry is an application of FE methods to a subfield of anthropology, known as geometric morphometrics that analyzes and compares the performance of different morphologies and morphospaces (distribution of morphologies) [85]. In fact, all studies on FE morphometry were conducted by Banabilh’s team on Malay participants, covering interests in the face, dental arch, and nasal airway morphology, as well as the cross-sectional area of the nasopharyngeal, oropharyngeal, and hypopharyngeal regions [43,44,45,46], as given in Table 3. Compared to controls, patients with OSA showed a substantial decrease in airway area [43], a significant difference in the bucco-submandibular region of the face [44], a smaller dental arch [45], and decreased size in the nasal valve and inferior turbinate [46]. The researchers also discovered that obesity may be one of the key confounding factors [44] and that the relationships with the geometric morphometrics warrant further investigations.

Studies that applied the standard finite element solver to solid mechanics are given in Table 4. This collection of papers focused on the development of constitutive models without emphasizing specific case scenarios. Uniform airflow pressure was applied on the inner wall of the airway as a boundary condition. Modeling the muscle activation was the major focus of four studies [57,59,62,63] and all of them employed Hill’s model [86], which divides the muscle into an active component with a contractile element and a passive component that contributes to elasticity. Kajee et al. [57] segmented the tongue into six compartments and evaluated the muscle activation model under gravity, even though only the results of the superior compartment were presented. Based on microhistology, tongue models were reconstructed and encoded with a muscular response to maintain the tongue position and to investigate the effects of gravity orientation and pressure loading on the fiber stretch [62,63]. Instead of the tongue muscle, Liu et al. [59] built the soft palate muscle using the constitutive Hill’s model encased in a neo-Hookean soft tissue model. They compared models with passive muscles, OSA models, and non-OSA models with active muscles driven by different activation stress settings. In addition to the deformation and displacement of the soft tissue, closing pressure is one of the parameters of interest, which is defined by the critical negative pressure that causes the soft palate to contact the pharynx wall [58,59].

On the other hand, Liu et al. [58] were interested in the surface tension of the mucosal lining between the soft palate and tongue. They segmented and connected the soft palate and tongue with cohesive elements. As a representation of humidity change in the airway, the influence of traction–separation strength on the closing pressure, soft palate displacement, and adhesion failure was explored. In addition, Carrigy et al. [47] attempted to determine Young’s moduli of muscle and adipose tissues by predicting the area–pressure correlations at the velopharynx and oropharynx cross-sections using FE simulations and matched the results to those of the in vivo experiments.

FSI is one of the main areas in interdisciplinary FE analysis that co-simulates the solid and fluid domains. One of the FSI models from Huang et al. [54] is illustrated in Figure 8. Their research indicated that FE analysis using just the fluid domain (i.e., computational fluid dynamics, CFD) would underestimate the prediction of the pressure difference and overestimate that of the negative flux [54]. In light of this, the research analyzed the air flux, pressure, and velocity field distribution of airflow simultaneously with the solid mechanics of the airway wall and soft tissues, which implicates airway collapse or narrowing due to wall collapse or soft palate/tongue obstruction. Airflow pressure/velocity exhibited a bi-directional connection with the cross-sectional area of the airway. Ilegbusi et al. [56] discovered that a supine position without dilator muscle activation decreased airway cross-sectional width at the tongue, epiglottis, and larynx levels. Dhaliwal et al. [52] reported that different patients had different sites of maximal collapse, including the soft palate, tongue base, and oropharynx. Moreover, the cross-sectional area of the airway was often illustrated with the displacement and deformation of the soft palate. Chen et al. [48] compared the displacement and deformation between the inspiration and expiration phases to demonstrate the one-way valve mechanism of the soft palate. Using the same technique, Sun et al. [68] demonstrated that OSA patients exhibited a bigger soft palate (i.e., hypertrophy) and displacement while breathing.

As given in Table 5, OSA was presumed by initiating an apnea breathing condition and compared to eupnea (i.e., normal breathing) by adjusting the boundary inlet air pressure [48,54], or by “paralyzing” the tongue (dilator) muscle [52,56,60,65]. The former could be accomplished by measuring pressure in vivo using a pressure transducer [48]. Specific techniques were presented to “paralyze” the tongue, i.e., to compare activation and inactivation of the dilator muscle, where collapsibility was defined by the changes in result between activation and inactivation [52]. Ilegbusi et al. [56] applied a force–time activation function for the genioglossus, in which the magnitude of the force was determined by matching the profile of the airway opening at the epiglottis level observed in trials of neurostimulation. Malhotra et al. [60] “activated” the dilator muscle by altering the elastic modulus of the soft tissue, whereas Rong et al. [65] utilized spring elements to simulate the function of muscle to resist deformation.

While some studies presented a normal surrogate (i.e., they generated the disorder by impoverishing the model of a “healthy” individual) [56,61], there were also patient-specific models [48,52,55,68]. Sun et al. [68] reconstructed models from both OSA and non-OSA individuals to account for morphological variations. Dhaliwal et al. [52] presented a case series of four OSA patients with varying degrees of apnea–hypopnea index (AHI), body mass index (BMI), and comorbidities to investigate if these variables contribute to the variability of simulation outcomes. Moreover, Mansour et al. [61] highlighted the measurement and application of heterogeneous upper airway pressure and simulated breathing during waking, rapid eye movement (REM) sleep, and non-REM sleep.

## 6. Discussion

The goal of this review is to identify and map the state-of-the-art computational biomechanics research on sleep-related topics, as well as its evidence gaps and directions for future research. While the focus of the review was on sleep medicine, we included sleep ergonomics since it is connected to sleep comfort, sleep health, and sleep disorders and implicates insomnia [78]. According to the categorization of ICSD, current research covered sleep-related breathing disorders, predominantly, OSA, and sleep-related movement disorders, including three articles on sleep bruxism. In fact, biomechanics may be less relevant to other areas of sleep medicine. An unknown source of chemosensitivity loss might be responsible for central sleep apnea [87]. Hypersomnia, narcolepsy, and restless legs syndrome may have neurological, genetic, or pharmacological causes [15], while individuals with parasomnia or rapid eye movement behavior disorder (RBD) may exhibit symptoms, such as confusional arousal, sleepwalking, and nightmare disorders [88], and may require sleep behavioral monitoring and prevent related injuries [89,90].

In this mapping review, OSA had the greatest number of computational biomechanical studies that utilized various FE techniques to assess various risk factors and mechanisms. FE morphometry addressed the association between anatomical variations and upper airway constriction. Constitutive models of muscle control of the tongue and soft palate have been built, and their effects, particularly under case-specific scenarios, have been examined in the FSI domain. Some findings have implications for the treatment plan and designs. Carrigy et al. [47] suggested that the equalized negative pressure drew the soft palate towards the tongue rather than the posterior pharyngeal wall, which offered insights into the designs of oral pressure therapy device. In addition, there were also FE studies on mandibular advancement device and quantitative data on the forward movement of the jaw and tongue to expand the airway space [26,91,92]. The detailed constitutive models of the palatal and tongue muscles might enable modeling of speech and swallowing [59] and facilitate a wider range of applications, especially in the field of dysphagia and aspiration, where in vivo experimentation and monitoring remain challenging and imprecise [93,94]. On the other hand, FE modeling of sleep bruxism aimed to investigate TMJ stress, and hence TMJ disorders under different risk factors, including tooth morphology, position, and direction of grinding. The symptoms of TMJ disorder included headache, earache, and facial pain. In addition to the articular disc of the TMJ, modeling the nerve pathway might give additional insights into the mechanism of pain and symptoms. Dental wear is another consequence of sleep bruxism [95] that necessitates further investigation into models of long-term energy absorption, dental material fatigue, and wear.

Sleep ergonomics could be a moderator/mediator of sleep deprivations (external and environmental factors) and interventions for sleep disorders. Defining good ergonomics and biomechanics for sleep may be among the most difficult unsolved problems. Under a supine posture, the research evaluated the sensitivity of pillow heights and mattress stiffnesses for an “ideal” spine curvature, which was assumed to be the upright standing position. A sagging spine due to insufficient regional support might induce discomfort, insomnia, and sleep-related musculoskeletal disorders [20]. Denninger et al. [51] attempted to customize the regional support of the body by integrating cells with varying stiffnesses. It was reported that floating on the Dead Sea promoted a relaxing recumbent posture. The buoyant force on sea-floating could self-adjust to different body parts, offering sufficient regional support load, which could keep the body joints and connections in a relatively strain-free condition. In fact, one of the targets of a comfortable bed is to facilitate the relaxation of muscles and intervertebral discs [96]. Hong et al. [53] reconstructed and analyzed the stress of the intervertebral discs; however, future work is required to model the passive or active stretch of the superficial and deep muscles of the spine, in addition to the initial conditions of strain-free states for the intervertebral discs to demonstrate relaxation. Musculoskeletal modeling has been used to estimate the muscle force or activation under different movements in various body parts, particularly the foot and spine [97,98,99,100], which may also be used to estimate that during sleep postures and to drive the FE models. A simulation of the spinal cord and arteries using FSI may also aid in revealing the mechanism underlying sleep-related cervicogenic headaches. Moreover, there was no research exploring side-lying positions in this review, and all evaluated supine posture. In fact, in vivo biomechanical investigations to analyze the spine curvature were more feasible for side-lying postures using optical techniques [101,102], ultrasound [103,104], and mechanical rollers [105,106] on the exposed back. The requirement for optimum side-lying spine curvature is more evident, in which the spine shall be aligned horizontally [20].

External validity and model validation were two of the most challenging aspects of relevant research. Reconstructing a single FE model and simulation takes strenuous effort and computational power, particularly when several anatomically detailed parts and complex simulations are involved (e.g., FSI). Based on the evidence map, most of the studies employed single-subject, case report, or case series designs, except for FE morphometry research (as cross-sectional studies), which needed substantially less processing power and time for analysis. Frequently, a single-subject design, claiming that the model subject was representative of the population, was utilized, whereas, in lieu of a patient-specific model, some studies attempted to use an impoverished “healthy” model as a surrogate representation of the pathology, thereby compromising external validity and generalizability [107]. While some studies accounted for external validity by creating and applying “statistically averaged” models and loading conditions, which were questionable whether the important features would be “flattened”. Despite the fact that some research accounted for external validity by developing and using “statistically averaged” models and loading conditions, it was dubious whether or not significant features would be “flattened”. Patient-specific models with corresponding patient-specific boundary and loading profiles are necessary [107]. While in silico models provide valuable information with the great translational potential to investigate intrinsic and extrinsic factors [107], in vivo studies and/or clinical trials are required to monitor patient-specific pathological and behavioral patterns and to help clinicians identify the underlying causes of the problem in real life [108,109].

Model validation is crucial for assessing how accurately the simulation represents the reality of interest, despite the paradox that computer simulation might not be necessary if a “real” experiment could be performed with the constraints on time, cost, feasibility, and complexity [107]. Dhaliwal et al. [52] took photographs inside the upper airway of the patients through an endoscopy to validate the collapse area of the simulation. Mansour et al. [61] acquired the upper airway airflow information with a pneumotachometer and pressure catheters to drive the model and validated the model predictions by visualizing the retropalatal lumen with a fiber-optic bronchoscope. Validation by means of lower-order data (not the primary outcome), alternative scenarios, scaled loading, or comparing with existing literature might facilitate an agreement between prediction and experimentation. Future studies may consider promoting translational potential and education/dissemination of findings using extended reality, such as virtual reality and augmented reality [110,111,112].

There were some limitations in this review. Limiting the search to papers written in English and published in journals indexed by the chosen electronic databases may introduce language and selection biases. Based on our pilot search, we also attempted to increase the specificity of the literature search by introducing a “NOT” operator for terms that often appeared in the incorrectly included articles. However, the approach might compromise the sensitivity of the literature search, thus we supplemented it with an ad hoc Google Scholar search for grey literature. The delimitations of this review were also worthy of note. Amputation, surgery, fracture, injury, and tumor may necessitate unique sleep considerations for affected individuals. Pressure sores are another kind of commonly related pathology, but they are not included in the ICSD [40]. Pressure sores may appear in different body locations (heels, sacrum, and shoulder) on different supports, such as bed, seat, and wheelchair [113,114,115], warranting another mapping review. On the other hand, we also did not include the scope of sleep-related interventions, which may include treatments for sleep disorders and sleep-related treatments for non-sleep disorders. The former may involve surgeries, oral appliances, and sleep posture therapy for OSA [116,117], while the latter may involve night braces/splints for other musculoskeletal conditions [118,119] or sleep posture and appliances for pregnancy [120]. Future reviews might consider scoping reviews on the modeling and simulation aspects to guide the improvement of research validity from a technical perspective.

## Figures and Tables

**Figure 1 bioengineering-10-00917-f001:**
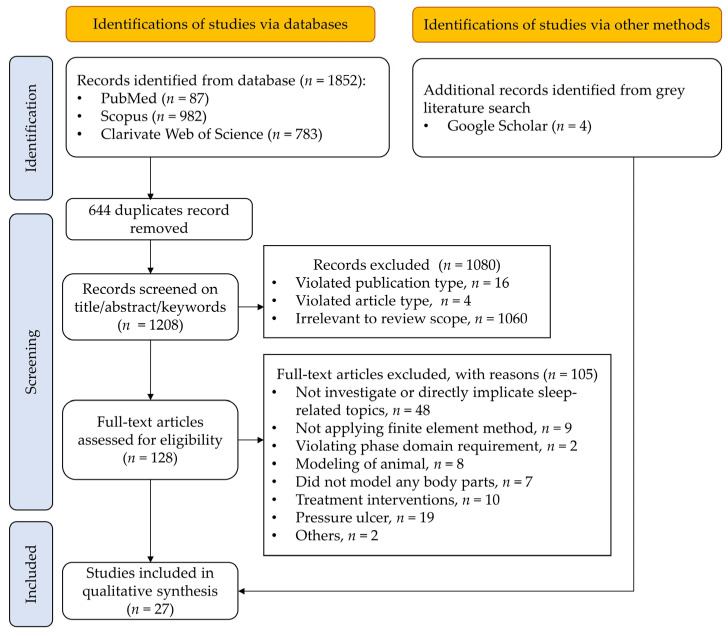
PRISMA flowchart for the structural review detailing database searches, studies screened, excluded, and retrieved.

**Figure 2 bioengineering-10-00917-f002:**
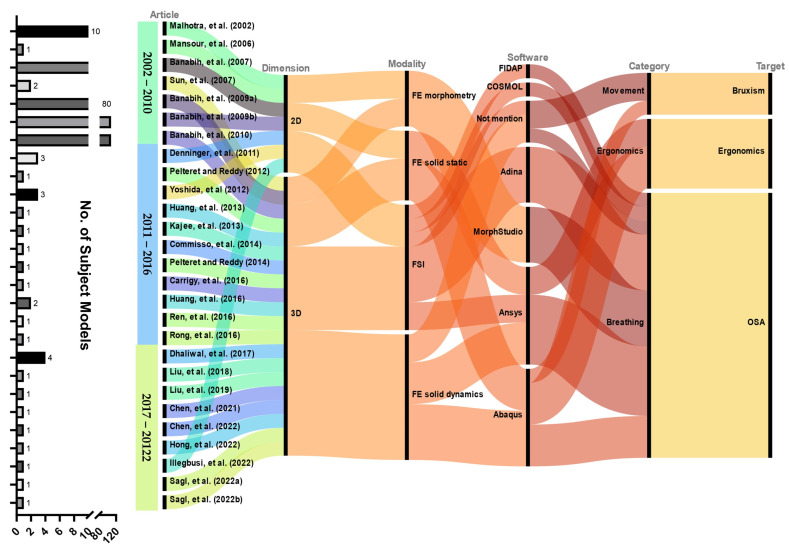
Sankey diagram mapping the categories and basic settings of the included studies [43,44,45,46,47,48,49,50,51,52,53,54,55,56,57,58,59,60,61,62,63,64,65,66,67,68,69,70]. FE, finite element; FSI, fluid–structure interaction; OSA, obstructive sleep apnea.

**Figure 3 bioengineering-10-00917-f003:**
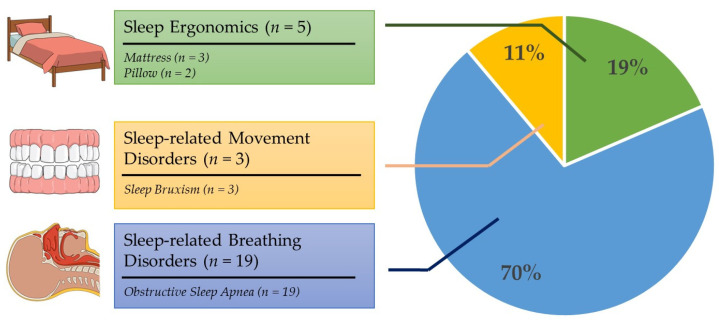
Categorization of included studies into sleep ergonomics (*n* = 5), sleep-related movement disorders (*n* = 3), and sleep-related breathing disorders (*n* = 19).

**Figure 5 bioengineering-10-00917-f005:**
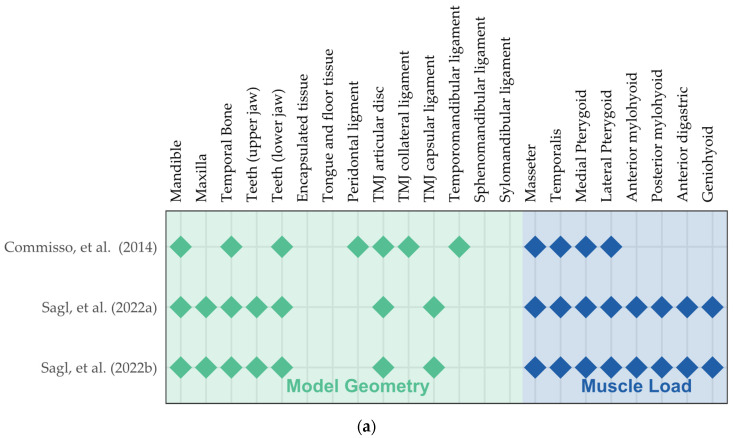
Evidence maps on modeling for the included studies related to (**a**) sleep bruxism [50,66,67] and (**b**) sleep ergonomics [49,51,53,64,70]. TMJ, temporomandibular joint.

**Figure 6 bioengineering-10-00917-f006:**
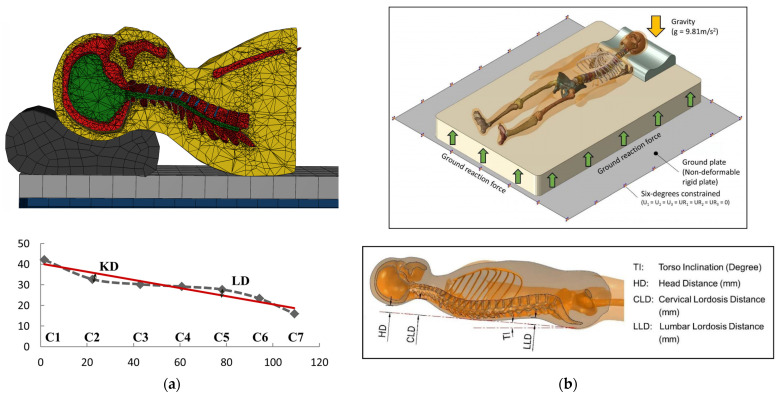
Computer models and definitions of outcome variables from (**a**) Ren et al. [64] and (**b**) Hong et al. [53] (source: [53,64], under Creative Commons Attribution License).

**Figure 7 bioengineering-10-00917-f007:**
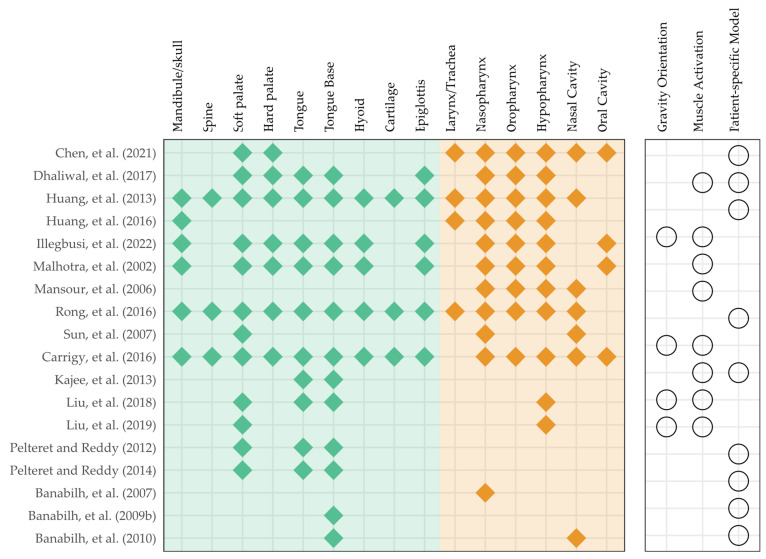
Evidence maps on modeling and features for included studies related to obstructive sleep apnea (OSA) [43,44,45,46,47,48,52,54,55,56,57,58,59,60,61,62,63,65,68].

**Figure 8 bioengineering-10-00917-f008:**
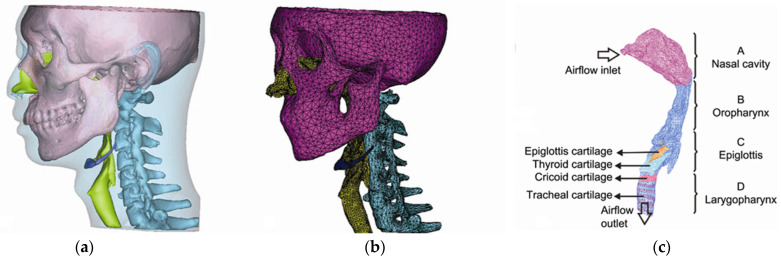
Illustration of the fluid–structure interaction finite element model with (**a**) model geometry, (**b**) model mesh in the solid domain, and (**c**) model mesh in the fluid domain (source: [54], under Creative Commons Attribution License).

**Table 1 bioengineering-10-00917-t001:** Design of the included studies related to sleep bruxism.

Study	Case Scenario(s)	Highlighted Feature(s)	* Primary Variant(s)	Outcome(s)
Commisso et al. [50]	Sustained and cyclic clenching	- Viscoelasticity of articular disc - Muscle loading pattern of the case scenarios	- Muscle activation level - Rate of muscle activation	Max/Min principal stress, max shear stress of TMJ articular disc
Sagl et al. [66]	Lateral bruxing	- Muscle activation derived from forward-dynamics	- Bruxing position - Bruxing inclination	Max bruxing force on molar and canine, VMS of TMJ articular disc
Sagl et al. [67]	Mediotrusive and laterotrusive bruxing	- Detailed tooth morphology - Simulation of bolus - Muscle activation derived from forward-dynamics	- Tooth position - Grinding direction - Bruxing force	TMJ loading, VMS of TMJ articular disc

TMJ, temporomandibular joint; VMS, von Mises stress. * Primary variant shall include the case scenarios.

**Table 2 bioengineering-10-00917-t002:** Design of the included studies related to sleep ergonomics.

Study	Case Scenario(s)	Highlighted Feature(s)	* Primary Variant(s)	Outcome(s)
Chen et al. [49]	Supine lying on pillow (without mattress)	- Verification of material constitutive model of Latex	Pillow height	VMS and displacement of headrest point, neck fossa and pillow
Denninger et al. [51]	Supine lying on mattress	- Customized mattress by numbers of hollow foam cubes with different hollow cavity dimensions - Established a simulation database for the design process	Foam cell design, configuration	Deformation of cube, spinal curvature (L4 to C7 levels)
Hong et al. [53]	Supine lying on mattress (with pillow)	- A comprehensive 3D body model - Unobstructive model validation on spinal alignment	Mattress stiffness	Peak pressure and contact area of occiput, cervical, scapula, buttocks, calves, and heel, VMS of IVD, spinal curvature (HD, CLD, TI, LLD)
Ren et al. [64]	Supine lying on pillow (with mattress)	- A comprehensive model of the cervical spine and skull with the brain	Pillow height	Peak pressure of the cranial and cervical regions, cervical spine alignment (CA, KD, LD)
Yoshida et al. [70]	Supine lying on mattress (without pillow)	- Linking FE results with perceived comfort and preference	Mattress stiffness	VMS of the cervical spine, relative sinking displacement between head and chest

CA, cervical angle; CLD, cervical lordosis distance; HD, head distance; KD, kyphotic distance; LD, lordotic distance; LLD, lumbar lordotic distance; TI, torso inclination; VMS, von Mises stress. * Primary variant shall include the case scenarios.

**Table 3 bioengineering-10-00917-t003:** Design of the included studies related to sleep apnea and applied FE morphometry.

Study	Population	OSA Group	Control Group	Outcome(s)
Banabilh et al. [43]	Malays	19 (13M, 6F)	19 (13M, 6F)	Airway area at nasopharyngeal, oropharyngeal, and hypopharyngeal regions
Banabilh et al. [44]	Malays	40	40	Facial morphology
Banabilh et al. [45]	Malays	54	54	Dental arch morphology
Banabilh et al. [46]	Malays	54 (32M, 22F)	54 (21M, 33F)	Nasal airway morphology

F, female; M, male; OSA, obstructive sleep apnea.

**Table 4 bioengineering-10-00917-t004:** Design of the included studies related to sleep apnea and applied standard FE-simulating solid mechanics.

Study	Highlighted Feature(s)	* Primary Variant(s)	Outcome(s)
Carrigy et al. [47]	- Determine a suitable set of material properties for computational models of pharyngeal mechanics - 3D anatomically detailed pharynx model	- Material properties of muscle and adipose tissue	Change of cross-sectional area at velopharyngeal and oropharyngeal regions per change in airway pressure, deformation of pharynx
Kajee et al. [57]	- Constitutive modeling of tongue - Detail modeling of tongue muscle compartments	- w/ and w/o gravity - w/ and w/o superior muscle activation	Tongue displacement
Liu et al. [58]	- Modeling of the adhesion between the soft palate and tongue by cohesive elements and traction-separation model	- Critical traction stress of cohesive element - Failure separation displacement of cohesive layer - Negative airway pressure	Displacement of soft palate, closing pressure, pressure level at adhesion failure
Liu et al. [59]	- Constitutive modeling of muscle activation for palatal muscle - Apply different EMG activation levels vs. negative pressure for healthy subjects and OSA patients	- Passive and active muscle models - OSA and non-OSA patients	Displacement of soft palate and closing pressure
Pelteret and Reddy [62]	- Active muscle contraction of tongue	- Orientation of gravity - Airway pressure - Individual tongue muscle activation - Position maintenance w/ the neural control model	Muscle activation response, displacement of tongue, muscle fiber stretch
Pelteret and Reddy [63]	- Active muscle contraction of tongue	- Orientation of gravity - Passive and active neural control model	Muscle activation response, displacement of the tongue, maximum principal stress of tongue, contractile stress of muscle fibers

3D, three-dimensional; EMG, electromyographic; OSA, obstructive sleep apnea; w/, with; w/o, without. * Primary variant shall include the case scenarios.

**Table 5 bioengineering-10-00917-t005:** Design of the included studies related to sleep apnea and applied FE co-simulation on fluid–structure interaction.

Study	Case Scenario(s)	Highlighted Feature(s)	* Primary Variant(s)	Outcome(s)
Chen et al. [48]	Inhalation and exhalation	- Patient-specific model - Measure pressure in the laryngeal cavity w/ a micro-pressure transducer catheter - Observe the one-way valve effect of the soft palate by high-speed dynamic MRI	Eupnea and apnea	Displacement and deformation (versus time) of soft palate, pressure distribution and velocity field of airflow in the upper airway cavity
Dhaliwal et al. [52]	Applying an inlet–outlet pressure difference	- Case series patient-specific models w/ validation	w/ and w/o muscle activation	Site of maximum collapse, degree of airway collapse
Huang et al. [54]	Inhalation and exhalation	- 3D anatomically detailed patient-specific model - Comparison between FSI and CFD technique	Eupnea and apnea FSI and CFD	Air flux, airflow pressure distribution of the airway cross-section, strain distribution of the sagittal cross-section of soft tissues
Huang et al. [55]	Arbitrary pressure	- Verification of constitutive modeling - Patient-specific	-	Airflow velocity distribution and streamline
Ilegbusi et al. [56]	Constant pharyngeal pressure to simulate inhalation	- Examine the positional (gravity) effects.	w/ and w/o dilator muscle activation, standing and supine position	Width of airway lumen at soft palate, tongue, epiglottis, and larynx levels; hyoid bone elevation; displacement of soft tissues; airflow rate and velocity distribution; airflow pressure distribution
Malhotra et al. [60]	Applying closing pressure values from existing literature	- Building models of 5 males and 5 females - Compare gender effects w/ respect to anatomy	w/ and w/o dilator muscle activation	Soft tissue deformation and displacement, closing pressure
Mansour et al. [61]	Respiratory cycle	- Visualized retropalatal airway by a fiber-optic scope - Measure airflow using a pneumotachometer and a pressure catheter	Wakefulness, non-REM sleep and REM sleep	Pharyngeal cross-sectional area
Rong et al. [65]	Inhalation and exhalation	- Modeling nerve control over the muscle by linear spring element on the anterior wall	w/ and w/o spring element	Deformation of airway cross-section, strain distribution of soft tissues in sagittal cross-section, airflow pressure and velocity distribution, airway resistance, flux
Sun et al. [68]	Inhalation and exhalation	- Patient-specific modeling	OSA and non-OSA model	Airflow pressure and velocity distribution, displacement of soft palate, airway resistance, flux

MRI, magnetic resonance imaging; OSA, obstructive sleep apnea; REM, rapid eye movement; w/, with; w/o, without. * Primary variant shall include the case scenarios.

## Data Availability

No new data were created or analyzed in this study. Data sharing is not applicable to this article.

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
