# Peer review of "Computational Biomechanics of Sleep: A Systematic Mapping Review"

_bioengineering, 2023, doi:10.3390/bioengineering10080917_

Round 1

Reviewer 1 Report

The paper is well-structured and well-written. The language is clear and the ideas.

Figures (2, 5a, 5b, and 7) show the references in a system different from those used in the text. The same in all the tables.

For coherence, all the expressions in Latin should be formatted in italics.

The abbreviation 3D was not defined. 

Author Response

We thank the reviewer for the time on the paper and the appreciation to our work. We also sincerely appreciate you revising our work on the attached file.

Figures (2, 5a, 5b, and 7) show the references in a system different from those used in the text. The same in all the tables.

We thank the reviewer for the reminder. We have revised Figure 5a, 5b, and 7 on the referencing system. We would like to retain the (author year) format for Figure 2 since the year information is essential in this Figure and we add the reference with the right format on the Legend to accommodate this situation.

 For coherence, all the expressions in Latin should be formatted in italics.

Thanks for the reminder and attached a file to indicate the non-italic words. We revise the Latin word, except those in-text citation (et al), which does not require italic according to the publisher’s citation guidelines. We will double confirm this issue with the editor.

 The abbreviation 3D was not defined.

Thanks for the reminder. The full term is supplemented accordingly.

Reviewer 2 Report

The paper is interesting and well written.  Only one suggestion: please mention eacu graph and figure in the main text. 

Author Response

The paper is interesting and well written.  Only one suggestion: please mention each graph and figure in the main text. 

Thanks for the reminder and appreciation. We double check on the cross-citation of the Figure and Tables and supplemented accordingly.

Reviewer 3 Report

The Manuscript is focused on an exciting topic. But problematic behaviours such as sleep are personal circumstances. Their underlying causes must be determined in vivo. In addition, sleep problems may occur in special situations such as illness. Computerized models may not give the correct result because the studies are insufficient.

Moderate editing of English language required.

Author Response

The Manuscript is focused on an exciting topic. But problematic behaviours such as sleep are personal circumstances. Their underlying causes must be determined in vivo. In addition, sleep problems may occur in special situations such as illness. Computerized models may not give the correct result because the studies are insufficient.

We concur with the limits of the study design of in silico research. We provide context in the discussion to emphasize this perspective.

“While in silico models provide valuable information with great translational potential to correlate intrinsic and extrinsic factors [107], in vivo studies and/or clinical trials are re-quired to monitor patient-specific pathological and behavioral patterns and to help clinicians identify the underlying causes of the problem in real life [108,109].”

Round 2

Reviewer 3 Report

A revised manuscript is acceptable.

By editorial decision: Moderate editing of English language required